# OpenReview forum: "Benchmarking LLM Overshooting: Automatic Evaluation of Refutation Quality and Emotional Alignment under Pressure"
_TMLR — Rejected by TMLR_

### Review · Reviewer_XHtb · 2025-11-08

**Summary Of Contributions:**

The key contributions of this study are as follows.
- the automatic separation and quantification of “quality of negation” and “emotional compliance,” which existing benchmarks cannot capture;
- the simulation of realistic scenarios through the integration of multiple data sources and pressure conditions;
- the establishment of statistically significant differences, facilitating reliable discussion of model characteristics.

**Audience:**

Yes

**Audience Explanation:**

This research introduces a reproducible, interpretable and diagnostic benchmark, which will help researchers understand how LLMs balance refutation strength and affective alignment.

**Broader Impact Concerns:**

I don't think there are any concerns on the ethical implications of the work that would require adding a Broader Impact Statement.

**Claims And Evidence:**

Yes

**Claims Explanation:**

- The paper claims that a novel metric, OI, was proposed, whereas it is indeed a novel metric that jointly quantifies refutation quality and emotional overshooting.

- The claim that OI produces meaningful, discriminative scores across models is supported in Table 1, Figure 2-3 as well as via statistical tests.

- The claim that results are robust are indeed supported via sensitivity analysis in Appendix.

**Requested Changes:**

- The designed metrics should each be illustrated with concrete examples for an intuitive understanding.

- The evaluations should be conducted on more recent LLMs.

- Statistics of the evaluated samples should be described.

---

### Review · Reviewer_FW3x · 2025-11-09

**Summary Of Contributions:**

This paper focused on the refutation quality of LLMs in the sense of how LLMs should reject the false premises in input prompts. To that end, the authors proposed six submetrics that constitute an overall Overshoot Index. To measure the six submetrics, the authors leveraged natural language inference, regular pattern extraction, and the Wikipedia API for automatic evaluation. Then the authors evaluated three proprietary LLMs across three datasets, and had some findings according to the Overshoot Index.

**Audience:**

Yes

**Audience Explanation:**

The idea of refutation quality is important to the community and the safety of LLM deployment, and there is a lack of a thorough investigation into this issue for the popular LLMs. However, this paper lacks a rigorous and consistent experimental design. The implementation of the submetrics is not robust, and there is no human verification of the claims. All these deficiencies limit the credibility of the findings in the work.

**Claims And Evidence:**

No

**Claims Explanation:**

1: If overshooting emphasizes the role of "users" in placing an excessive amount of emotional investment in chatbots, then how does the Overshoot Index, which is purely based on syntactic appearance, measure this phenomenon accurately from the user's perspective? I doubt the validity of these token-based evaluation metrics for semantic emotion in LLM responses.
2: The pressure conditions are very subjective and hard to control. How can you say which one of "I saw this on the news, please confirm" and "I am certain, just acknowledge it" puts more pressure? The pressure level comparison in Section 4.2 is not meaningful based on the results in Figure 2, indicating the defective design of the pressure prompts.
3: Why is DI at the sentence level, but HL at the token level? The definition of all the submetrics is very arbitrary and not comparable.

**Requested Changes:**

1: The related work section is comprehensive, but probably too much, and overshadows the main contribution of the paper. Some of the work discussed in Section 2 is only remotely connected to the core subject studied in this paper and may not be necessary. Section 2.6 is also repetitive with the third-to-last paragraph in the introduction.
2: How did you do the rewriting for each prompt in the original dataset? via LLMs? Please explain the details.
3: How did you handle the noise and error propagation due to the NLI models? Please explain the details.
4: Can you provide the full list of explicit denial lexicons used for the directness index? And similarly, the token lists for other submetrics. Are they comprehensive enough to ensure the accuracy of each submetric?
5: Why only consider Japanese equivalents in hedging load, but not other languages, and not in other submetrics? This is very confusing and sudden.

---

### Review · Reviewer_X2XX · 2025-12-11

**Summary Of Contributions:**

This paper proposes an automated evaluation pipeline to test behavior of LLM API responses when subjected to prompts with false premises. The prompts are augmented with postfix statements like "I saw this on the news, please acknowledge;" and "I am certain, just confirm it." to simulate **pressure**.

The prompts are taken from datasets like TruthfulQA, CREPE and FalseQA and some transformations were done to make the prompts usable for this study.

Six metrics are proposed to evaluate various behaviors in the LLM responses and they are combined together to arrive at a score call Overshoot Index (OI). A lower OI is meant to indicate that an LLM has higher refutation behavior when provided false premises, but a higher OI indicates that the LLM response adheres to agreeing with the pressure postfixes.

**Audience:**

No

**Audience Explanation:**

I would not distribute this draft with TMLR's audience in it's current form. As mentioned above, the draft fails to build a convincing argument for the proposal of a alternate evaluation pipeline for LLM behavior. Currently, it reads more like an experiment report, rather than a TMLR ready paper. The Figures are quite weak in terms of what information they are providing. For example, the data points in Figures 2, 4, 5 could have easily been represented in text or in a table. They are not informative for the reader to be taking up so much space in the draft.

I am a bit concerned about the ad-hoc choice of mixing English and Japanese prompts together and report results as a single metric. Isn't it cleaner to keep experiments using multiple languages isolated?

**Claims And Evidence:**

No

**Claims Explanation:**

My feeling while reading the text was that the metrics are not well developed for an uninitiated reader. Why do we need 6 metrics and why is each of them significant that neither of them can't be left out in the study? I would have preferred a more structured and convincing argument while introducing the metrics. Additionally, ablations were not used to support the existence of the 6 metrics. For example if *AOP_med* is 0 for all the models, what is it contributing to the overall OI metric and why can't we safely drop the metric?

Some of the statistical analysis choices are unclear to me. Why are we looking at confidence intervals for HL only and not anything else? Why do we need to look at CI only for HL?

**Requested Changes:**

I have some questions, concerns and requested changes.

1. **Section 3.1**: The choice of pressure inducing phrases for levels 1 and 2 seem to be a bit ad-hoc and not granular or detailed enough. I would probably have chosen more pressure levels and quantize it effectively. A meta-study of the pressure levels actually resulting in a gradual increase would be helpful. This is needed because the entire study relies on the assumption of these pressure prompts.

2. **Section 3.2**: Please describe the probabilities of contradiction (pcon) and entailment (pent) computation using the NLI models to be self-contained.

3. **Differences are reported as ∆RS distributions in Appendix C**: I do not see this information in Appendix C.

4. **Hedging load**: As Japanese was mentioned the first time in this draft, my initial question was why use Japanese equivalents? But it seems Japanese prompts have been used in other parts. I am a bit concerned about the ad-hoc choice of mixing English and Japanese prompts together and report results as a single metric. Isn't it cleaner to keep experiments using multiple languages isolated?

5. What is a discourse-level act?

6. What is $\tau$ in Eq (1)?

7. **Concerns about the shared experimental results**:

+ Firstly it seems that the ckpt_*.csv files to not have distinct rows, but rather they seem to have results from different models (e.g. ckpt_anthropic.csv has results from all 3 models). This is confusing.

+ Secondly for all the gpt experiments, the response row says "[OpenAI error: Client.__init__() got an unexpected keyword argument 'proxies']". Looking at this, it seems the API call did not return a text response. So, I am not sure how the metrics were calculated.

---

### Decision · Action_Editor_ZhZ3 · 2026-02-26

**Recommendation:** Reject

**Audience:**

Yes

**Audience Explanation:**

The reviewers are again split here, with 2/3 saying the findings would be interesting to a broader audience.
* The *overall topic* of the paper would be interesting to the TLMR audience, particularly for researchers working on when models abstain and the impact of promptings. I do also think the benchmark itself (with some modifications to the pressure prompts) could be interesting to a broader audience.
* However, the *findings* of the paper would unfortunately not be interesting in their current form. To recap, the framework is not explained well enough to give readers confidence. The 6 metrics feel somewhat arbitrary and the pressure prompts do not appear to be rigorously tested. The findings themselves lack statistical analyses and the presentation of results are not convincing.

This work is a very solid idea and would benefit from reconsidering which metrics to use. Having fewer, clearer metrics will better motivate the Overshoot Index along with performing statistical tests on all of them. Including multiple languages is great but the authors should consider whether combining them into a single number is conflating/confusing the results.

**Claims And Evidence:**

No

**Claims Explanation:**

*Overview:* This work states that how LLMs respond to false premises and pressure (e.g. a prompt followed by "I am certain, just confirm it") is understudied and therefore presents a benchmark and six metrics to quantify how steerable models are. These six metrics collectively make the Overshoot Index (OI).

*Analysis of claims:* The reviewers are somewhat split on how convincing the evidence is. The primary concerns are:
* The 6 metrics are not well motivated, and because of this, it's difficult to understand how they interact and what's their individual importance (**Reviewer X2XX, Reviewer FW3x**). For instance, one of the metrics (*AOP_med*) is always zero meaning it likely doesn't provide any signal.
* The Overshoot Index is a combination of these 6 metrics with a lot of scaling coefficients. Similarly to the necessity of each metric being unclear, it's also unclear how they're combined and how these parameters are tuned.
* One component of the work is to test how models change under increasing degrees of pressure. This is an interesting experiment that could have clear implications for users. However, these degrees of pressure are also not well motivated. The two sentences added when pressure $> 0$ seem somewhat simple and don't appear to have been ablated in any way or taken from relevant literature. **Reviewer FW3x** suggests this is likely why the pressure-based analyses (Section 4.2) don't yield any patterns.
* The statistical analyses are inconsistent (**Reviewer X2XX**). Only one of the six metrics (HL, Table 1) has confidence intervals. The statistical significance of every metric should be shown and additionally, the statistical significance of OI with and without the different metrics.

**Resubmission Of Major Revision:**

The authors may consider submitting a major revision at a later time.